bioengineering/biomaterials/biomechanics

electron spun membranes, wettability, synovial fluid, periprosthetic osteolysis, surface energy

**Author for correspondence:**
Lukas Capek
e-mail: lukas.capek@tul.cz

# The wettability of electron spun membranes by synovial fluid

Ales Hrouda[1,3], Radek Jirkovec[1], Petra Hamrikova[2], Maarten Vanierschot[3], Kathleen Denis[3] and Lukas Capek[1]

[1]Faculty of Textile Engineering, TU Liberec, Liberec 46117, Czechia
[2]Department of Forensics Pathology, Regional Hospital in Liberec, Husova 10, 46001 Liberec, Liberecký, Czechia
[3]Department of Mechanical Engineering, KU Leuven, Leuven, Flanders, Belgium

KD, 0000-0002-6492-9607; LC, 0000-0003-3950-0646

Aseptic loosening due to periprosthetic osteolysis has been accepted as one of the leading causes of revision procedures in patients with previous joint arthroplasty. Recently, several strategies for suppression of osteolysis were proposed, mostly based on biological treatment such as mitigation of chronic inflammatory reactions. However, these biological treatments do not stop the debris migration but only reduce the inflammatory reaction. To address this shortcoming, we propose the concept of ultrahigh molecular weighted polyethylene particles filtration storage by electrospun membranes. Firstly, the surface tension of synovial fluid (SF) is obtained by use of a pendant droplet. Secondly, the contact angle of the electrospun membranes wetted by two different liquids is measured to obtain the free surface energy using of the Owens–Wendt model. Additionally, the wettability of electrospun membranes by SF as a function of technology parameters is studied.

## 1. Introduction

Electrospun materials have shown a large scale of biomedical applications in the last decade and therefore still draw increasing attention in this field [1–4]. Among the outstanding benefits of such materials, the following must be mentioned: high biocompatibility, tunable solubility, size and structural variability, straightforward chemical functionalization and, on a large scale, productivity. In regard of functionalization and filtration properties, the wettability of the interface between biological liquid and the membrane must be known [5,6]. This information determines whether the membrane is hydrophilic or hydrophobic in nature and thus aids in determining the

**Figure 1.** The embodiment of a membrane-covered hip implant.

applications in which it can most readily be employed. Therefore, understanding the exact wettability characteristics will greatly assist with any membrane future applications.

In this study, we have focused on a concept of a new implant using an electrospun membrane for the prevention of ultrahigh molecular weighted polyethylene (UHMWPE) wear-induced osteolysis (figure 1). This is the process by which prosthetic debris mechanically released from the surface of prosthetic joints induces an immune response that favours bone catabolism, resulting in loosening of the prosthesis with eventual failure or fracture [7]. Recently, several strategies for the suppression of osteolysis were proposed mostly based on biological treatment such as the mitigation of chronic inflammatory reactions [8–10]. However, these biological treatments do not stop the debris migration but only reduce the inflammatory reaction. To address this shortcoming, we propose the concept of UHMWPE debris filtration storage by electrospun membrane. The electrospun membrane would envelope the implant and thus capture and store the wear particles. We suppose that the synovial fluid (SF) can flow through the membrane. This concept aims to reduce the number of particles that flow into the bone and as such reducing the osteolysis development. As a first step, the wettability of the developed membranes by SF is studied.

SF is the liquid in the synovial cavity and is secreted by the synovial membrane. Its function is to reduce friction between the articular cartilages of the synovial joint during movement. It has an essential role in the physiological function of all large joints in the human body. It is supposed that the intracapsular hip joint volume is lower than 10 ml and has 2.7 ml of SF in it [11,12]. Even though there are studies focused on the viscosity of SF, there is nearly no knowledge about surface tension and its wettability with various materials [13–17].

It was shown that the wettability of non-woven membranes changes according to a different topography structure of the non-woven materials and that it can also be influenced by different surface modifications [18–20]. On the other hand, the surface modification is another technology step which might influence and complicate the whole fabrication process. Thus, only process parameters of the electrospinning, such as towing velocity, number of layers and final thermal treatment process, are considered further, not only the nature of this technology, but also in regard to possible use and filtration applications in medicine. Each of those parameters can affect the fibre morphologies and therefore the wettability and surface tension of SF, but to the best of the authors' knowledge, none of them was studied in the past in the case of SF and electrospun membranes interaction.

# 2. Material and methods

## 2.1. Biological sample retrieval

SF samples from hip joints were obtained from six human cadavers (70 ± 7 years) at the Department of Forensic Pathology (Regional Hospital of Liberec, Czechia) during forensic autopsies with post-mortem intervals ranging between 6 and 48 h under cooled conditions. Informed consent was obtained from the donor's next of kin, in line with the approval of the ethics committee and in accordance with the Declaration of Helsinki. The study was performed in accordance with the Czech Human Tissue Act 2004. None of the cadavers were affected with hip injury or had undergone hip arthroplasty.

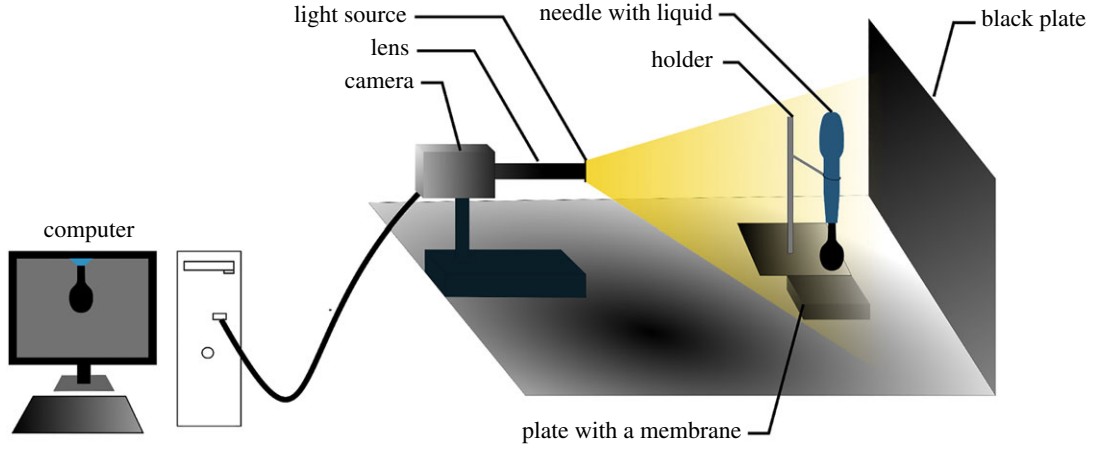

**Figure 2.** The scheme of the pendant droplet measurement set-up.

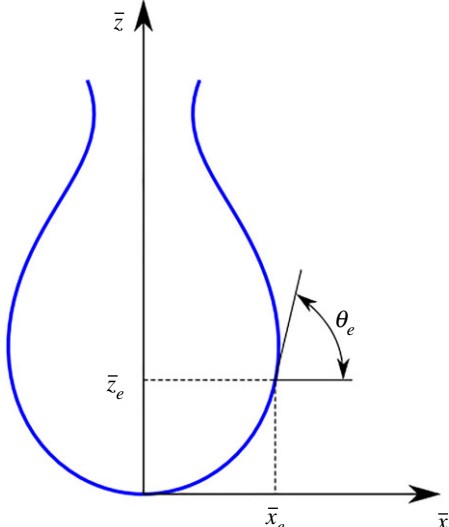

**Figure 3.** The pendant droplet with the dimensionless coordinates.

## 2.2. Surface tension measurement

Various methods for the measurement of the surface tension can be used [21–25]. One of them is called the pendant droplet [25], and it is a common method that calculates the surface tension coefficient of a liquid by image processing [26]. Figure 2 shows a detailed scheme of the measurement set-up that was built based on the study of Raj *et al.* [27]. This set-up was used to measure the surface tension coefficient of the SF. The liquid is sucked by a pipette and afterwards placed into a holder to fix the position of the pipette in all directions. The tip of the pipette is placed 7 mm above the plate with a membrane. The applied pressure on the pipette is slightly increased, and its values and thus all the stages of the droplet evolution can be recorded by a camera, the Image Source DFK 33UX250 (Image Source Ltd, UK) with a Navitar 12X lens (Navitar, Inc., USA). The pendant droplet was recorded with the software NIS-Elements 5.10 (Nikon Inc., Japan).

The shape of the pendant droplet (figure 3) can be numerically described by a set of differential equations that are derived from the Young–Laplace equation [28].

$$\frac{\partial \bar{x}}{\partial \bar{s}} = \cos \theta, \tag{2.1}$$

$$\frac{\partial \bar{z}}{\partial \bar{s}} = \sin \theta \tag{2.2}$$

and

$$\frac{\partial \theta}{\partial \bar{s}} = 2 - \beta \bar{z} - \frac{\sin \theta}{\bar{x}}, \tag{2.3}$$

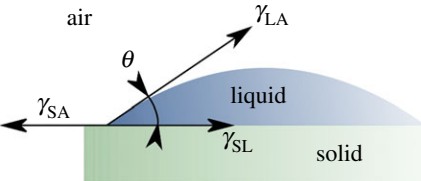

**Figure 4.** The contact angle measurement.

where $\bar{x} = x/R_0$ is the dimensionless coordinate in the $x$-direction, $\bar{z} = z/R_0$ is the dimensionless coordinate in the $z$-direction, $\bar{s} = s/R_0$ is the length of the droplet and $\theta$ is the angle, $R_0$ is the initial radius of curvature, $\beta$ is the shape factor coefficient and is defined as $\beta = (g \cdot \Delta\rho R_0^2)/\gamma$, with $\Delta\rho$ the difference between the fluid density and the air density, $g$ is the gravitational constant ($9.81\ \mathrm{m\,s^{-2}}$) and $\gamma$ is the surface tension coefficient.

The density of the used fluid is considered to be known, as does the gravitational constant $g$, and thereby in the expression for $\beta$, two unknowns remain, i.e. $R_0$ and $\gamma$. Fordham [29] proposed a calculation of the shape factor coefficient $\beta$ as a function of the ratio $\sigma$ between $d_s$ and $d_e$. Hansen & Rødsrud [30] adjusted the previous calculation of the shape factor coefficient $\beta$.

$$\beta = 0.12836 - 0.7577\sigma + 1.77132\sigma^2 - 0.54263\sigma^3 \tag{2.4}$$

Although the shape parameter value can now be calculated, the surface tension coefficient remains unknown. Therefore, Hansen & Rødsrud [30] derived an expression for the ratio $d_e/2R_0$ as follows:

$$\frac{d_e}{2R_0} = 0.99787 + 0.1987\beta - 0.07342\beta^2 + 0.347083\beta^3 \tag{2.5}$$

From equation (2.5), the initial radius of curvature can be obtained, and the surface tension coefficient is calculated as follows:

$$\gamma = g \cdot \frac{R_0^2 \Delta\rho}{\beta}. \tag{2.6}$$

Additionally, the free surface energy (FSE) of the solid was studied. Hence, the sessile droplet method (figure 4) was used to obtain the contact angle between the liquid and solid phase. Moreover, by using two liquids with a known surface tension coefficient, the value of the FSE can be calculated. The sessile droplet method was recorded with the same set-up as for the pendant droplet (figure 2). The post-processing of the result was done in the software NIS-Elements 5.10.

$$\gamma_{SA} = \gamma_{SL} + \gamma_{LA} \cos \theta. \tag{2.7}$$

In Young's equation [31] (equation (2.7)), $\gamma_{SA}$ is the FSE of the solid, $\gamma_{LA}$ is the surface tension of the liquid phase and $\gamma_{SL}$ is the interaction energy between the solid and liquid phase. For this interaction, theoretical models were derived. In this study, the model describing the solid–liquid interaction, the Owens–Wendt model [32], was used,

$$1 + \cos \theta = 2\sqrt{\gamma_{SD}}\frac{\sqrt{\gamma_{LD}}}{\gamma_{LA}} + 2\sqrt{\gamma_{SH}}\frac{\sqrt{\gamma_{LH}}}{\gamma_{LA}}, \tag{2.8}$$

where $\theta$ is the contact angle, $\gamma_{SD}$ is the polar solid surface free energy coefficient, $\gamma_{SH}$ is the dispersive free solid surface energy coefficient, $\gamma_{LD}$ is the polar liquid surface energy coefficient and $\gamma_{LH}$ is the dispersive liquid surface energy coefficient,

$$\gamma_{LA} = \gamma_{LD} + \gamma_{LH} \tag{2.9}$$

The FSE $\gamma_{SA}$ is calculated from equations (2.8) and (2.9). In these equations, there are two unknowns, and thus the measurement needs to be done for two liquids. In this study, the used liquids were water and glycerol. For these liquids, the polar and dipolar values of surface tension are known, and their values are shown in table 1. The FSE was calculated in the software Matlab (The MathWorks, Inc., USA).

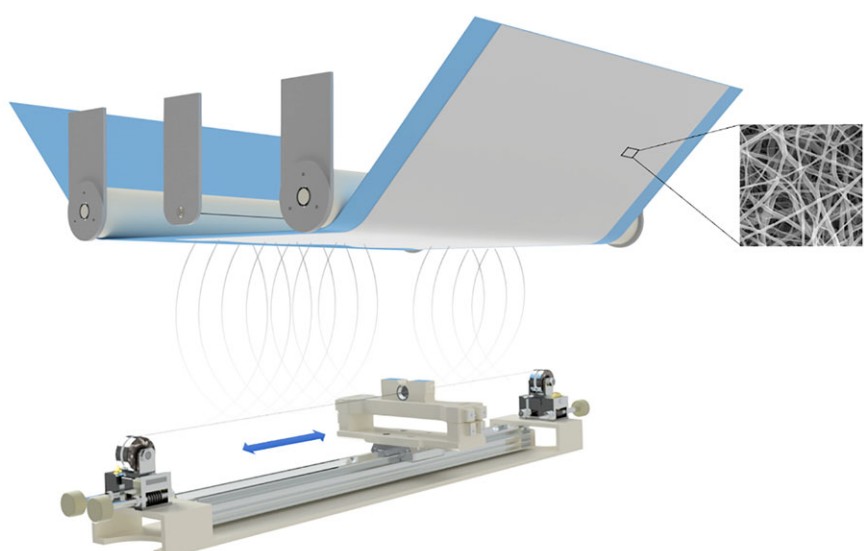

**Figure 5.** The scheme of the electrospinning process.

**Table 1.** The properties of the used liquid, water and glycerol [33].

| liquid | $\gamma_{LD}$ (mN m$^{-1}$) | $\gamma_{LH}$ (mN m$^{-1}$) | $\gamma_L$ (mN m$^{-1}$) |
|---|---|---|---|
| water | 21.8 | 51 | 72.8 |
| glycerol | 33.6 | 29.7 | 63.3 |

## 2.3. Membrane sample preparation

For the preparation of the electrospun membranes, polyamide 6 (PA6, Ultramid B27, BASF, Germany) was used. The solvent system consisted of acetic and formic acids (Penta Chemicals, Czechia). The polymer solution of PA6 was prepared in a concentration of 12 wt%. The solvent system consisted of the above-mentioned acids in a ratio of 2 : 1. The solution was mixed in the magnetic stirrer at room temperature for 24 h until the polymer was fully dissolved. The prepared solution was electrospun by using a Nanospider NS 1WS500U (Elmarco, Czechia) with a wire electrode. The voltage was set at 50 kV, and the voltage of the opposite collector's electrode was −10 kV. The distance between the electrode and collector was set at 180 mm. The process of electrospinning was done at a temperature of 22°C and a relative humidity of 40%. Different towing velocities (5–20 mm min$^{-1}$) were used to investigate the influence on the contact angle. The electrospinning was done on a polypropylene base Spunbond fabric. Afterwards, the electrospun layers were laminated by use of a hydraulic press HVL 15.2 (Pracovní stroje Teplice, Czechia), the layers were connected by a constant press force kept at 15 kN and the lamination temperature was a changing parameter (180–200°C). The electrospun membranes were analysed by scanning electron microscopy (SEM, Tescan Vega3, Czechia). Before SEM, the membranes were gold plated on Quorum Q150R ES. Afterwards, these samples were placed in Tescan Vega3, and the pictures were taken under a voltage of 20 kV. The SEM images were used for determining the fibre diameter by analysing in the software ImageJ [34]. The thickness of the membrane was measured with an Elcometer 456 (Garmin, Czechia).

Figure 5 is a scheme of electrospinning using the Nanospider device. The polymer solution is poured into a cartridge, which applies the polymer solution to the string electrode by its movement. Due to the high voltage, Taylor cones form on the electrode and subsequently form fibres. The fibres are carried to the counter electrode, where they are captured by the Spunbond layer.

# 3. Results

## 3.1. The surface tension coefficient of the SF

Firstly, the measurements of the pendant droplet were performed with water. Hence, the surface tension coefficient is known (72.5 mN m$^{-1}$) [33]. The resultant surface tension coefficient was obtained with a

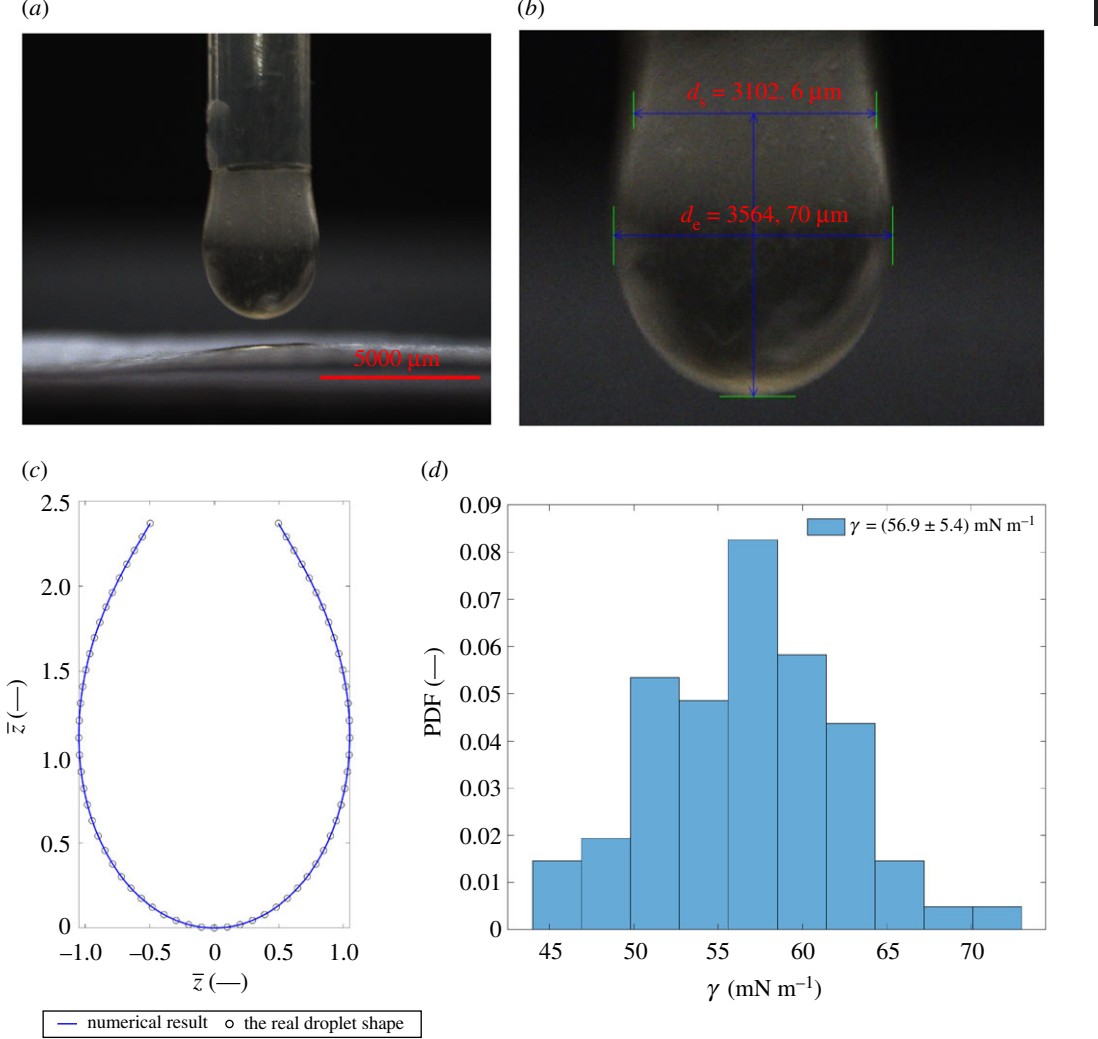

**Figure 6.** (a) The pendant droplet measurement. (b) The measured diameters $d_e$ and $d_s$. (c) The comparison of the numerical solution with the measured droplet. (d) The distribution of the surface tension coefficient of SF. PDF: probability distribution function.

value of 77.5 mN m$^{-1}$, with an error smaller than 10%. Therefore, the used methodology and set-up were suitable to be used for the SF measurements.

The SF was retrieved from six patients with an average age of 70 years, and for each patient, 13 measurements were performed. A measurement sample picture is shown in figure 6a. The diameters $d_e$ and $d_s$ were measured (figure 6b) during the post-processing phase. By applying equations (2.5)–(2.7), the surface tension coefficient $\gamma$ of the SF was obtained. The numerical simulation of the shape of the droplet [25,35] (figure 6c) of the SF was obtained, and it is in good agreement with the real droplet shape [36,37]. The initial conditions for the numerical solution were $\bar{x} = \bar{z} = \theta = 0$ [29].

Table 2 shows the calculated surface tension coefficient for each patient. Figure 6d shows the detailed distribution of the surface tension coefficient of SF that was retrieved from six patients. The distribution of the surface tension coefficients approximates a normal distribution with a mean diameter of 56.9 mN m$^{-1}$, and the standard deviation of 5.37 mN m$^{-1}$.

## 3.2. The electrospun membranes

For the electrospinning, the towing velocity of the Spunbond was set at 5, 10, 15 and 20 mm min$^{-1}$, and the obtained material characteristics are shown in table 3. Firstly, for the non-thermally influenced samples, the free surface tension was calculated, and it is shown in figure 7. Yalcinkaya et al. [38,39] have shown that the lamination pressure can influence the structure of the membranes and so the contact angle. However, the lamination force was kept constant (15 kN), and the lamination

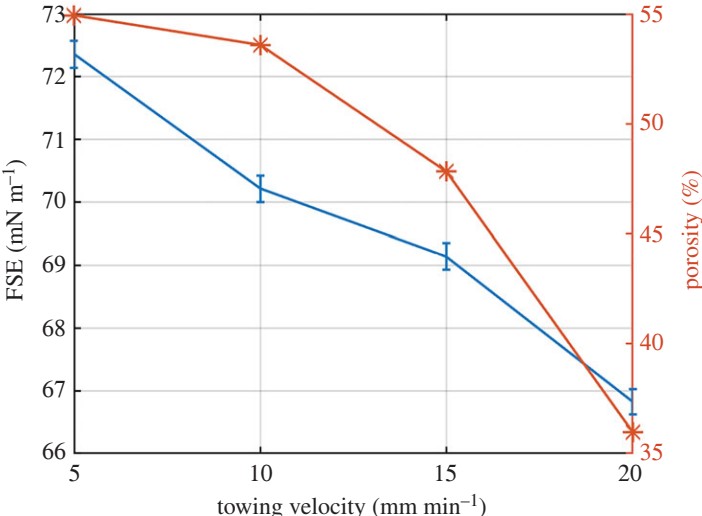

**Figure 7.** FSE and porosity for electrospun membranes with different towing velocities.

**Table 2.** Obtained surface tension coefficients for six patients, where F = female and M = male.

| patient no. | gender | age (years) | $\gamma$ (mN m$^{-1}$) |
| --- | --- | --- | --- |
| 1 | F | 73 | 61.3 ± 5.2 |
| 2 | M | 74 | 53.3 ± 4.7 |
| 3 | M | 61 | 58.8 ± 4.0 |
| 4 | F | 71 | 56.6 ± 2.6 |
| 5 | M | 58 | 59.7 ± 4.6 |
| 6 | M | 84 | 51.7 ± 5.4 |
| average | | 70.1 ± 8.6 | 56.9 ± 5.4 |

**Table 3.** The virgin samples of PA6 for different towing velocities.

| towing velocity (mm min$^{-1}$) | fibre diameter (µm) | thickness (µm) | surface density (g m$^{-2}$) | porosity (%) |
| --- | --- | --- | --- | --- |
| 5 | 0.16 ± 0.04 | 20.9 ± 2.1 | 12.7 ± 0.4 | 46.19 |
| 10 | 0.19 ± 0.08 | 16.2 ± 1.7 | 8.6 ± 0.5 | 53.27 |
| 15 | 0.24 ± 0.08 | 8.5 ± 0.9 | 5.2 ± 0.2 | 46.19 |
| 20 | 0.24 ± 0.10 | 5.3 ± 1.0 | 3.7 ± 0.1 | 38.49 |

temperature was changed. Although the towing velocity of 20 mm min$^{-1}$ has resulted in membranes with the lowest FSE values, these membranes were taken for the next lamination process. This towing velocity might be considered as the worst case scenario, and the authors wanted to investigate the effect of lamination on the surface structure and also on the wettability. The layers were laminated during a lamination time of 3 s, and the lamination temperature was ranging from 180 to 200°C with an increase of 10°C per sample.

The fibre diameter was measured in the software ImageJ, and the number of measured fibres was considered to be at least 100. Figure 8 shows the fibre diameter distribution for a towing velocity of 20 mm min$^{-1}$. The porosity of the electrospun membranes was calculated as follows:

$$\epsilon = 1 - \frac{\rho_{\text{bulk}}}{\rho_{\text{mat}}},$$ (3.1)

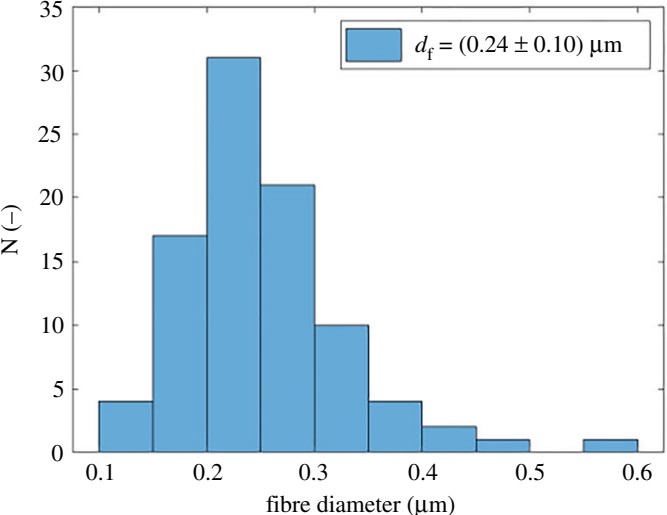

**Figure 8.** The fibre diameter distribution for the towing velocity of 20 mm min$^{-1}$. The average fibre diameter had a value of 0.24 µm. The total number of measured fibres was 100.

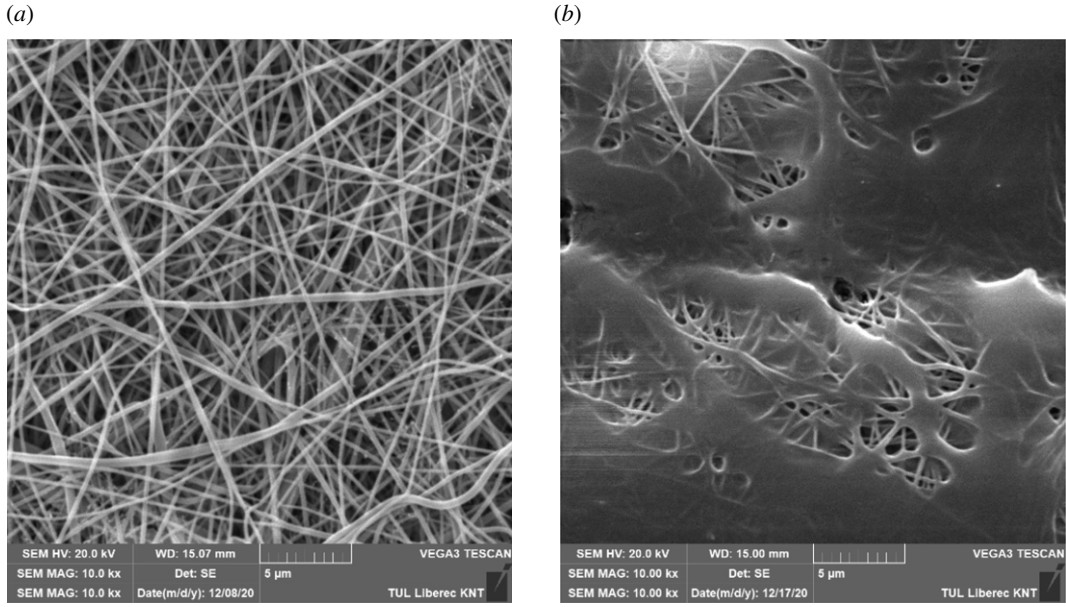

**Figure 9.** The temperature effect on the surface treatment. (*a*) Surface after lamination with a lamination temperature of 180°C. (*b*) Surface after lamination with a temperature of 205°C.

where $\rho_{bulk}$ is the density of the membrane and $\rho_{mat}$ is the density of the fibres. In this case, the latter is the density of the used material (PA6) and is equal to 1130 kg m$^{-3}$ [40].

The towing velocity affects the FSE (figure 7), one can see that the FSE does not change significantly. The difference between the highest and lowest FSE is 6 mN m$^{-1}$, and one can note that with the decreasing porosity, the FSE also decreases.

The range of the lamination temperatures has been chosen from the empirical experience of the lamination process. The lamination for temperatures lower than 180°C does not influence the structure enough (figure 9*a*). However, the membranes start to create a foil for temperatures higher than 200°C as it follows from the melting point of PA6 [41] (figure 9*b*).

Table 4 shows the affected parameters of the membrane for a towing velocity of 20 mm min$^{-1}$ by adding the lamination process into the production procedure. By increasing the number of layers, the porosity increases. However, the influence of the lamination process on the porosity is not clear. One can expect that with an increasing number of layers, the surface density and the thickness of the membrane increase.

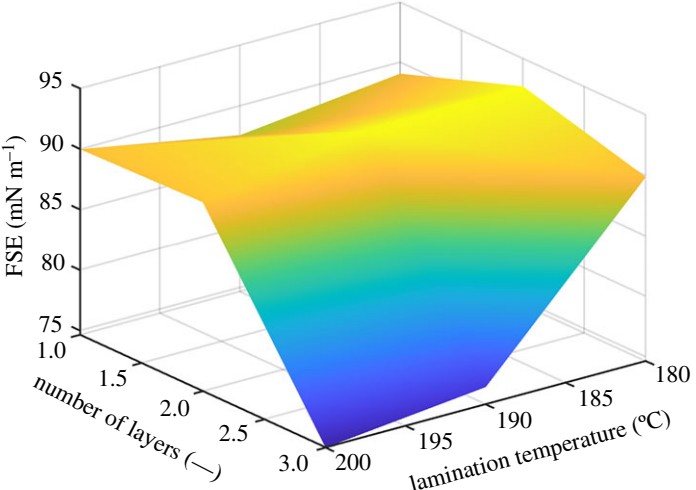

**Figure 10.** The FSE for the constant towing velocity of 20 mm min$^{-1}$, constant lamination time of 3 s, the temperature range of 180–200°C and the number of layers that varied from one to three layers.

**Table 4.** The parameters of the membranes after the lamination process that was applied on the membrane produced by a towing velocity of 20 mm min$^{-1}$.

| lamination temperature (°C) | number of layers (—) | thickness (μm) | surface density (g m$^{-2}$) | porosity (%) |
|---|---|---|---|---|
| 180 | 1 | 4.8 ± 0.7 | 4.3 ± 0.2 | 20.95 |
| | 2 | 11.4 ± 0.9 | 7.8 ± 0.5 | 39.84 |
| | 3 | 18.0 ± 1.0 | 10.7 ± 0.4 | 47.21 |
| 190 | 1 | 4.9 ± 0.6 | 4.0 ± 0.1 | 28.7 |
| | 2 | 13.0 ± 1.0 | 8 ± 0.3 | 45.58 |
| | 3 | 19.0 ± 1.1 | 11.6 ± 0.6 | 45.64 |
| 200 | 1 | 5.1 ± 0.8 | 4.6 ± 0.3 | 19.17 |
| | 2 | 11.9 ± 1.2 | 8.0 ± 0.8 | 40.78 |
| | 3 | 18.9 ± 1.3 | 12.5 ± 0.4 | 41.55 |

The highest FSE has been obtained for the temperature of lamination of 180°C, and the number of layers to be two (figure 10). From figure 10, it follows that the number of layers and the temperature of lamination play important roles for the wetting.

## 3.3. Wetting the electrospun membranes by synovial fluid

Figure 11a shows the measurement of the contact angle between the SF and the electrospun membrane produced for a towing velocity of 20 mm min$^{-1}$. Figure 11b shows that with increasing towing velocity and decreasing porosity, the contact angle increases. Moreover, the contact angle raises as it was expected from figure 7 to be increasing. The FSE for a towing velocity of 20 mm min$^{-1}$ has shown the lowest value. Therefore, a towing velocity of 20 mm min$^{-1}$ was chosen as an optimal parameter to investigate the influence of the lamination process.

The process parameters showed to have a big influence on the wettability of the membrane (figure 12). The optimal temperature for the lamination in the case of PA6 is 190°C, and the number of layers is one to two. As follows from the figure, for lower FSE, the contact angle is expected to be higher [42]. The lamination showed an influence on the wettability by the fluid [39,40]. For a lamination temperature of 200°C and higher, a membrane foil creation started to occur, and this influences the wettability badly. As can be seen from figure 12, the optimal candidate has a lamination temperature of 190°C, and the number of layers is two.

(*a*)

(*b*)

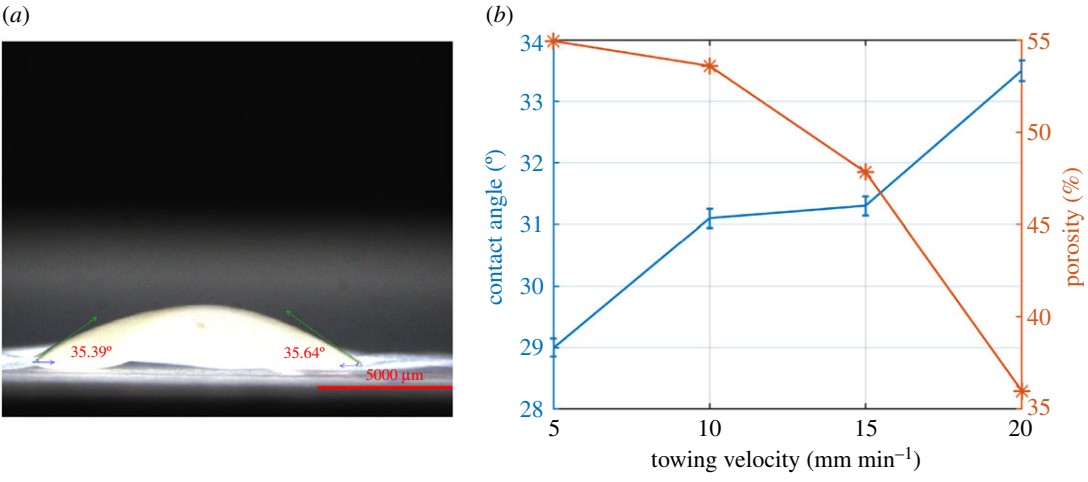

**Figure 11.** (*a*) The contact angle measurement for the synovial fluid for the membrane of 20 mm min$^{-1}$. (*b*) The contact angle for synovial membrane for different towing velocities.

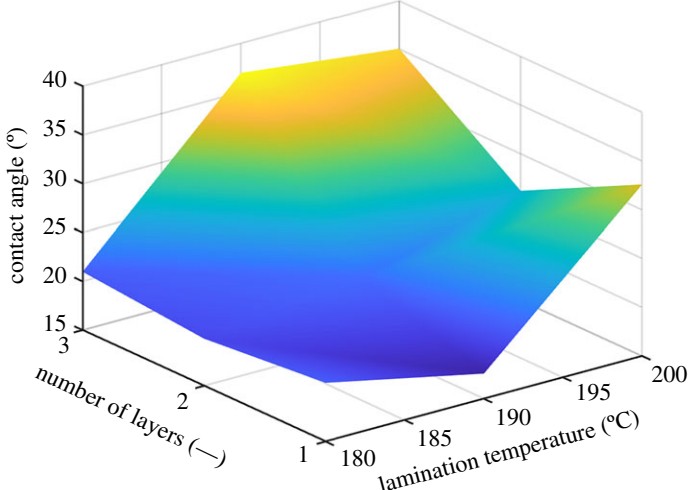

**Figure 12.** The effect of the number of layers and lamination temperature for a towing velocity of 20 mm min$^{-1}$ and a lamination duration of 3 s.

## 4. Discussion

SF is normally a transparent, viscous liquid found inside synovial joints that functions as a biomechanical lubricant and as a medium for metabolites and soluble signalling factors. Its composition varies between individuals and between healthy joints and arthritic or diseased joints. In the case of a total hip prosthesis present, the SF serves also as a fluid in which UHMWPE debris is present. The most wear debris from an artificial joint has a spherical or subspheroidal shape, and a small amount has an unbroken plate structure, with an average diameter around 1 µm [43,44]. Recently, several strategies for suppression of osteolysis were proposed by biological treatment or searching alternative materials for articulation in total joint replacement [8,9,45]. However, these treatments do not stop the debris migration but only reduce the inflammatory reaction or the amount of wear. In this study, we have focused on a concept of a new implant using an electrospun membrane for the prevention of UHMWPE wear-induced osteolysis. One of the new membrane's key functionality factors is the wettability of the interface between biological liquid and membrane.

Surface tension is a property of the surface of a liquid that allows it to resist an external force. It was shown that the liquid surface tension affects the filter performance [46–48]. In our study, the surface tension of SF was measured by a pendant drop test. The average value was calculated to be 56.9 ± 5.4 mN m$^{-1}$. Jeleniewicz *et al*. [49] studied the surface tension by the Wilhelmy plate method on

patients with seronegative spondyloarthropathies and rheumatoid arthritis. They have found values between 42 and 48 mN m$^{-1}$. They pointed out that measuring the surface tension could be suitable for differential diagnosis of the type of joint disease. Hills & Butler [50] obtained a surface tension coefficient of SF of 50 mN m$^{-1}$ on a dog model. It can be seen that the values measured in our study are 14% higher than those reported in literature. This difference might be due to the different test groups in the Jeleniewicz's study, which were spondyloarthropathies and rheumatoid arthritis patients group, and in the current study, these were elderly. Since the studied group of SF was of an elderly age, some quantitative changes could be found. It is known that the viscosity of SF is varying with age [51]. On the other hand, there is no up-to-date knowledge on how the surface tension of SF is influenced by this parameter. This needs to be examined further.

The authors suppose that wettability is an important factor influencing not only the functionality of electrospun membranes but also their filtration efficiency. The non-thermally influenced electrospun membranes have shown different contact angles which were related to different porosities. The towing velocity of 20 mm min$^{-1}$ has shown the worst case scenario in the range between 5 and 20 mm min$^{-1}$, as it has the greatest contact angle. The selected process parameters of the lamination of electrospun membranes, such as the number of layers and the lamination temperature, showed a big influence on the wettability of the material. The optimal temperature for the lamination in the case of PA6 can be considered to be 190°C with the number of layers being one or two.

## 5. Conclusion

This work focused on the interface interaction between SF and electrospun membranes with different production parameters, i.e. the towing velocity, number of layers and lamination temperature. The aim of the study was to obtain the optimal set of parameters that might be later used in the filtration of the wear particles to prevent osteolysis development.

Firstly, the surface tension coefficient of the SF was obtained by the use of the pendant droplet method. Secondly, the FSE of electrospun membranes with the different production parameters mentioned above was obtained by measuring the contact angle for two different liquids with known surface tension coefficients, i.e. water and glycerol. The FSE of the electrospun membranes was then calculated using the Owens–Wendt model. At last, the contact angle between the SF and the electrospun membranes was measured. From the contact angle and FSE of solid, the optimal candidate for the electrospun membrane was produced with the following production parameters: a lamination temperature of 180°C, a towing velocity of 20 mm min$^{-1}$ and one or two layers.

Data accessibility. Image analyses results files are available from the Dryad Digital Repository: http://doi.org/10.5061/dryad.59zw3r28p [52].

Authors' contributions. All authors gave final approval for publication and agreed to be held accountable for the work performed therein.

Competing interests. We declare we have no competing interests.

Funding. We received no funding for this study.

Disclaimer. The manuscript has not previously been published in print or electronic form and is not under consideration by any other publication.

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
