## [Peer Review File · Royal Society Open Science]

Review History

RSOS-210892.R0 (Original submission)

Review form: Reviewer 1

Is the manuscript scientifically sound in its present form?

Yes

Are the interpretations and conclusions justified by the results?

Yes

Is the language acceptable?

Yes

Do you have any ethical concerns with this paper?

No

Have you any concerns about statistical analyses in this paper?

No

Recommendation?

Accept with minor revision (please list in comments)

Comments to the Author(s)

Dear Authors,

in your interesting manuscript, the following points should be added/changed to further improve it:

- Introduction: The idea to add a nanofibrous layer to reduce friction-related release of material from the lower macroscopic material sounds a little bit weird. Please explain a little bit better what exactly the additional nanofiber mat should do and what it has to do with the synovial fluid.
- The last sentence of the introduction sounds strange, as if no process parameters had ever been studied. Please make this clearer, too.
- Surface tension measurement: Here it should be mentioned that you plan to measure the surface tension of the synovial fluid, else the reader might wonder where the nanofiber mats enter the game.
- above Eq. 4: "ratio between d_s and d_e ", ditto below Eq. 4
- Eq. 5: d_e or D_e ?
- Membrane sample preparation: "the electrospun layers were laminated by use of a hydraulic press" - laminated with what? Or just pressed? If so, at which temperature and pressure?
- Electrospun membranes: "temperature was ranging from 180 to 200°C with an increase of 10°C" - an increase per sample or per second or ...?
- Wetting the electrospun membranes: "Figure 11b shows that with increasing towing velocity and decreasing porosity." - this sentence is incomplete.
- page 9: "56.90 ± 5.37 mN/m" - there are brackets missing, it must be (56.9 ± 5.4) mN/m, else the average doesn't have a unit. And standard deviations have max. 2 significant digits.
- Fig. 1: Is it possible to add a scale bar in the magnification glass?
- Fig. 2: Please try making the inset descriptions a little bit larger, they are hard to read.
- Fig. 6a: The scale bar description is not readable.
- Fig. 6d: Please add brackets to γ , as mentioned above. And if there is no unit, it is not necessary to show this by brackets with nothing in.
- Fig. 7: Please add error bars.
- Fig. 8: Please add brackets. The standard deviation has too many digits, and the average must have the same accuracy, so it should simply be (0.24 ± 0.07) μm
- Figs. 10, 12: It should be (°C), not (degC), and the non-existence of units shouldn't be mentioned in this way.
- Fig. 11a: the right angle is misleading, is it possible to exchange it by the real contact angle?
- Fig. 11b: Please add error bars.
- Tables 2, 3: Please reduce the significant digits of the standard deviations to max. 2 and the accuracy of the average accordingly.
- Table 4: Please add standard deviations.

Review form: Reviewer 2

Is the manuscript scientifically sound in its present form?

No

Are the interpretations and conclusions justified by the results?

No

Is the language acceptable?

Yes

Do you have any ethical concerns with this paper?

Yes

Have you any concerns about statistical analyses in this paper?

Yes

Recommendation?

Major revision is needed (please make suggestions in comments)

Comments to the Author(s)

The authors constructed surface free energy structures by electrospinning to alleviate chronic inflammatory response. The data and conclusions of the paper have certain interesting and significance. However, there are some problems in analysis and data.

1. The residue of filtration performance is an important cause of inflammation. The author emphasizes the surface structure, but does not explain the filtration performance. The filtration performance, voids and other data of the membrane shall be provided. Surface and wetting are surface characteristics. Is the corresponding filtration related?
2. Do you consider the thickness, porosity, surface properties and affinity of electrospinning membrane materials?
3. Young-Laplace equation has existed for a long time, and the corresponding calculation of material surface tension also has corresponding instructions. The key is what is the difference between the surfaces of porous materials? What is the novelty of the author's research?
4. As the receiving fabric of electrospinning, what is the basic structure of the fabric, such as specification, thickness, gram weight, plain, twill and non-woven fabric? Fabric structure has a great influence on electrospun membranes.
5. In the electrospinning process, the fiber diameter affects the porosity. Does the corresponding surface tension have an effect? Should the method be used to prove?
6. Why only choose the advancing speed of 15 mm/min for spinning? The relationship between fiber diameter and surface tension should be increased. At the same time, the surface morphology after hot pressing and the hot pressing of advancing speed should be improved. In addition, whether the corresponding pressing temperature is too low, because the melting point of PA6 is obviously greater than 200 °C, how can there be a molten layer without softening, or just hot pressing the non-woven layer, what does it have to do with the electrospinning layer?
7. Page 8, Line 12. Is the conclusion wrong? Is there no direct relationship between surface tension and porosity?
8. Why is the electrospun membranes hydrophilic? What is the calculation method of porosity?
9. It is suggested that the author further refine the research work, especially the corresponding characterization data, to increase the persuasion of the paper. It can be considered to focus on the relationship between the structure and properties of the obtained electrospun membranes.

Decision letter (RSOS-210892.R0)

Dear Dr Capek:

Title: The wettability of electron spun membranes by synovial fluid
Manuscript ID: RSOS-210892

The editor assigned to your manuscript has now received comments from reviewers. We would like you to revise your paper in accordance with the referee and Subject Editor suggestions which can be found below (not including confidential reports to the Editor). Please note this decision does not guarantee eventual acceptance.

Please submit your revised paper before 22-Oct-2021. Please note that the revision deadline will expire at 00.00am on this date. If we do not hear from you within this time then it will be assumed that the paper has been withdrawn. In exceptional circumstances, extensions may be possible if agreed with the Editorial Office in advance. We do not allow multiple rounds of revision so we urge you to make every effort to fully address all of the comments at this stage. If deemed necessary by the Editors, your manuscript will be sent back to one or more of the original reviewers for assessment. If the original reviewers are not available we may invite new reviewers.

Yours sincerely,
Dr Ellis Wilde
Publishing Editor, Journals

On behalf of the Subject Editor Professor Anthony Stace and the Associate Editor Professor Chaohua Cui.

RSC Associate Editor
Comments to the Author:
(There are no comments.)

RSC Subject Editor
 Comments to the Author:
 (There are no comments.)

Reviewers' Comments to Author:

Reviewer: 1

Comments to the Author(s)

Dear Authors,

in your interesting manuscript, the following points should be added/changed to further improve it:

- Introduction: The idea to add a nanofibrous layer to reduce friction-related release of material from the lower macroscopic material sounds a little bit weird. Please explain a little bit better what exactly the additional nanofiber mat should do and what it has to do with the synovial fluid.
- The last sentence of the introduction sounds strange, as if no process parameters had ever been studied. Please make this clearer, too.
- Surface tension measurement: Here it should be mentioned that you plan to measure the surface tension of the synovial fluid, else the reader might wonder where the nanofiber mats enter the game.
- above Eq. 4: "ratio between d_s and d_e ", ditto below Eq. 4
- Eq. 5: d_e or D_e ?
- Membrane sample preparation: "the electrospun layers were laminated by use of a hydraulic press" - laminated with what? Or just pressed? If so, at which temperature and pressure?
- Electrospun membranes: "temperature was ranging from 180 to 200°C with an increase of 10°C" - an increase per sample or per second or ...?
- Wetting the electrospun membranes: "Figure 11b shows that with increasing towing velocity and decreasing porosity." - this sentence is incomplete.
- page 9: "56.90 ± 5.37 mN/m" - there are brackets missing, it must be (56.9 ± 5.4) mN/m, else the average doesn't have a unit. And standard deviations have max. 2 significant digits.
- Fig. 1: Is it possible to add a scale bar in the magnification glass?
- Fig. 2: Please try making the inset descriptions a little bit larger, they are hard to read.
- Fig. 6a: The scale bar description is not readable.
- Fig. 6d: Please add brackets to γ , as mentioned above. And if there is no unit, it is not necessary to show this by brackets with nothing in.
- Fig. 7: Please add error bars.
- Fig. 8: Please add brackets. The standard deviation has too many digits, and the average must have the same accuracy, so it should simply be (0.24 ± 0.07) μm
- Figs. 10, 12: It should be (°C), not (degC), and the non-existence of units shouldn't be mentioned in this way.
- Fig. 11a: the right angle is misleading, is it possible to exchange it by the real contact angle?
- Fig. 11b: Please add error bars.
- Tables 2, 3: Please reduce the significant digits of the standard deviations to max. 2 and the accuracy of the average accordingly.
- Table 4: Please add standard deviations.

Reviewer: 2

Comments to the Author(s)

The authors constructed surface free energy structures by electrospinning to alleviate chronic inflammatory response. The data and conclusions of the paper have certain interesting and significance. However, there are some problems in analysis and data.

1. The residue of filtration performance is an important cause of inflammation. The author emphasizes the surface structure, but does not explain the filtration performance. The filtration performance, voids and other data of the membrane shall be provided. Surface and wetting are surface characteristics. Is the corresponding filtration related?
2. Do you consider the thickness, porosity, surface properties and affinity of electrospinning membrane materials?
3. Young-Laplace equation has existed for a long time, and the corresponding calculation of material surface tension also has corresponding instructions. The key is what is the difference between the surfaces of porous materials? What is the novelty of the author's research?
4. As the receiving fabric of electrospinning, what is the basic structure of the fabric, such as specification, thickness, gram weight, plain, twill and non-woven fabric? Fabric structure has a great influence on electrospun membranes.
5. In the electrospinning process, the fiber diameter affects the porosity. Does the corresponding surface tension have an effect? Should the method be used to prove?
6. Why only choose the advancing speed of 15 mm/min for spinning? The relationship between fiber diameter and surface tension should be increased. At the same time, the surface morphology after hot pressing and the hot pressing of advancing speed should be improved. In addition, whether the corresponding pressing temperature is too low, because the melting point of PA6 is obviously greater than 200 °C, how can there be a molten layer without softening, or just hot pressing the non-woven layer, what does it have to do with the electrospinning layer?
7. Page 8, Line 12. Is the conclusion wrong? Is there no direct relationship between surface tension and porosity?
8. Why is the electrospun membranes hydrophilic? What is the calculation method of porosity?
9. It is suggested that the author further refine the research work, especially the corresponding characterization data, to increase the persuasion of the paper. It can be considered to focus on the relationship between the structure and properties of the obtained electrospun membranes.

Author's Response to Decision Letter for (RSOS-210892.R0)

See Appendix A.

RSOS-210892.R1 (Revision)

Review form: Reviewer 1

Is the manuscript scientifically sound in its present form?

Yes

Are the interpretations and conclusions justified by the results?

Yes

Is the language acceptable?

Yes

Do you have any ethical concerns with this paper?

No

Have you any concerns about statistical analyses in this paper?

No

Recommendation?

Accept with minor revision (please list in comments)

Comments to the Author(s)

Dear Authors,

After revising your manuscript according to my comments, I would now suggest publishing the manuscript as is.

Review form: Reviewer 2

Is the manuscript scientifically sound in its present form?

Yes

Are the interpretations and conclusions justified by the results?

Yes

Is the language acceptable?

Yes

Do you have any ethical concerns with this paper?

Yes

Have you any concerns about statistical analyses in this paper?

Yes

Recommendation?

Accept as is

Comments to the Author(s)

The author has modified as required. The corresponding figures 1 and 2 should be modified appropriately, and the corresponding color matching of the atlas can be more beautiful. At the same time, tables 1 and 2 should put the data on one page.

Decision letter (RSOS-210892.R1)

Dear Dr Capek:

Title: The wettability of electron spun membranes by synovial fluid
Manuscript ID: RSOS-210892.R1

It is a pleasure to accept your manuscript in its current form for publication in Royal Society Open Science. The chemistry content of Royal Society Open Science is published in collaboration with the Royal Society of Chemistry.

Yours sincerely,
Dr Ellis Wilde
Publishing Editor, Journals

On behalf of the Subject Editor Professor Anthony Stace and the Associate Editor Professor Chaohua Cui.

RSC Associate Editor
Comments to the Author:
Please consider to address the technical issues raised by the reviewer #2 during the proofreading stage.

RSC Subject Editor
Comments to the Author:
(There are no comments.)

Reviewer(s)' Comments to Author:
Reviewer: 1
Comments to the Author(s)
Dear Authors,

after revising your manuscript according to my comments, I would now suggest publishing the manuscript as is.

Reviewer: 2
Comments to the Author(s)
The author has modified as required. The corresponding figures 1 and 2 should be modified appropriately, and the corresponding color matching of the atlas can be more beautiful. At the same time, tables 1 and 2 should put the data on one page.

Appendix A

Letter to reviewers:

We thank you for the interesting and helpful remarks to our article. The contents of the paper were revised according to the specified suggestions. The altered sections are in red. **We suppose that now the article is well prepared for your prestigious journal.**

Reviewer #1:

Comment 1:

Introduction: The idea to add a nanofibrous layer to reduce friction-related release of material from the lower macroscopic material sounds a little bit weird. Please explain a little bit better what exactly the additional nanofiber mat should do and what it has to do with the synovial fluid.

Answer 1:

The electrospun membrane would envelope the implant and thus capture, store the wear particles. We suppose that the synovial fluid can flow through the membrane. This concept aims to reduce the number of particles that flow into the bone and thus also reduce the osteolysis development. Currently, we are working on the filtration process of UHMWPE particles within total hip replacement by an electrospun membrane [e.g. Hrouda A. et al. Statistical prediction of PM2.5 filtration applied in a case of electrospun membranes with a pore size distribution obtained from SEM, Nanofibers, Applications and Related Technologies 2021, Istanbul or Hrouda et al. Macroscale simulation of the filtration process of porous media based on statistical capturing models, Separation and Purification Technology 266(7):118577]. This is the key factor. Working on this research we have found that the wettability of electrospun membranes is not described elsewhere and this information is missing in our work.

Changes 1:

The primary idea was explained and added into the introduction.

Comment 2:

The last sentence of the introduction sounds strange, as if no process parameters had ever been studied. Please make this clearer, too.

Answer 2:

Yes, you are right. Thanks for this comment.

Changes 2:

The last sentence of the introduction was revised according to your suggestion.

Comment 3:

Surface tension measurement: Here it should be mentioned that you plan to measure the surface tension of the synovial fluid, else the reader might wonder where the nanofiber mats enter the game.

Answer 3:

Yes, you are right. Thanks for this comment.

Changes 3:

The corresponding paragraph was revised according to your suggestion: This setup was used to measure the surface tension coefficient of the synovial fluid.

Comment 4:

above Eq. 4: "ratio between d_s and d_e ", ditto below Eq. 4

Answer 4:

Indeed, it is a typo.

Changes 4:

The corresponding sentence was corrected.

Comment 5:

Eq. 5: d_e or D_e ?

Answer 5:

Indeed, it is a typo.

Changes 5:

The corresponding equation was corrected.

Comment 6:

Membrane sample preparation: "the electrospun layers were laminated by use of a hydraulic press" - laminated with what? Or just pressed? If so, at which temperature and pressure?

Answer 6:

Thank you for this comment. The electrospun layers were connected by a constant press force kept at 15 kN. The lamination temperature was parametric (changing).

Changes 6:

This information was added into the corresponding methodology section.

Comment 7:

Electrospun membranes: "temperature was ranging from 180 to 200°C with an increase of 10°C" - an increase per sample or per second or ...?

Answer 7:

The layers were laminated during a lamination time of 3 seconds and the lamination temperature was ranging from 180 to 200°C with an increase of 10°C **per sample**.

Changes 7:

The corresponding sentence was corrected.

Comment 8:

Wetting the electrospun membranes: "Figure 11b shows that with increasing towing velocity and decreasing porosity." - this sentence is incomplete.

Answer 8:

Yes, you are right. Thanks for this comment.

Changes 8:

Figure 11a shows the measurement of the contact angle between the synovial fluid and the electrospun membrane produced for a towing velocity of 15 mm/min. Figure 11b shows that with increasing towing velocity and decreasing porosity **the contact angle increases**.

Comment 9:

page 9: " 56.90 ± 5.37 mN/m" - there are brackets missing, it must be (56.9 ± 5.4) mN/m, else the average doesn't have a unit. And standard deviations have max. 2 significant digits.

Answer 9:

Ok, thank you for this remark. The sentence was revised according to your suggestion.

Changes 9:

The sentence was revised according to your suggestion: The average value was calculated to **(56.9 ± 5.4) mN/m**.

Comment 10:

Fig. 1: Is it possible to add a scale bar in the magnification glass?

Answer 10:

Yes, indeed. Thank you for this suggestion.

Changes 10:

The scale bar was added into figure 1.

Comment 11:

Fig. 2: Please try making the inset descriptions a little bit larger, they are hard to read.

Answer 11:

Thank you for this comment.

Changes 11:

The inset was made larger.

Comment 12:

Fig. 6a: The scale bar description is not readable.

Answer 12:

Thank you for this comment. The scale bar was adapted.

Changes 12:

The scale bar was made larger.

Comment 13:

Fig. 6d: Please add brackets to gamma, as mentioned above. And if there is no unit, it is not necessary to show this by brackets with nothing in.

Answer 13:

NO

Changes 13:

The brackets were added into the figure description.

Comment 14:

Fig. 7: Please add error bars.

Answer 14:

NO

Changes 14:

The graph was changed to an error bar chart.

Comment 17:

Fig. 8: Please add brackets. The standard deviation has too many digits, and the average must have the same accuracy, so it should simply be $(0.24 \pm 0.07) \text{ um}$

Answer 17:

NO

Changes 17:

It was changed to (0.24 ± 0.07) μm .

Comment 18:

Figs. 10, 12: It should be ($^{\circ}\text{C}$), not (degC), and the non-existence of units shouldn't be mentioned in this way.

Answer 18:

NO

Changes 18:

The units were corrected.

Comment 19:

Fig. 11a: the right angle is misleading, is it possible to exchange it by the real contact angle?

Answer 19:

NO

Changes 19:

The right contact angle was corrected.

Comment 20:

Fig. 11b: Please add error bars.

Answer 20:

NO

Changes 20:

The error bars were added

Comment 21:

Tables 2, 3: Please reduce the significant digits of the standard deviations to max. 2 and the accuracy of the average accordingly.

Answer 21:

NO

Changes 21:

It has been changed accordingly.

Comment 22:

Table 4: Please add standard deviations.

Answer 22:

NO

Changes 22:

The standard deviations were added

Reviewer #2:

The authors constructed surface free energy structures by electrospinning to alleviate chronic inflammatory response. The data and conclusions of the paper have certain interesting and significance. However, there are some problems in analysis and data.

Comment 1:

The residue of filtration performance is an important cause of inflammation. The author emphasizes the surface structure, but does not explain the filtration performance. The filtration performance, voids and other data of the membrane shall be provided. Surface and wetting are surface characteristics. Is the corresponding filtration related?

Answer 1:

Thank you for this comment. This presented study is a part of our research focused mainly on the filtration process of UHMWPE particles within total hip replacement by an electrospun membrane. The performance of the electrospun membrane during the filtration process can be find in a following conference paper “Hrouda A. et al. Statistical prediction of PM2.5 filtration applied in a case of electrospun membranes with a pore size distribution obtained from SEM, Nanofibers, Applications and Related Technologies 2021, Istanbul]. This is the key factor of this research. Working on this topic we have found that the wettability of electrospun membranes is non described elsewhere and this information is missing in our work.

Changes 1:

We have added the information about the primary idea into the introduction.

Comment 2:

Do you consider the thickness, porosity, surface properties and affinity of electrospinning membrane materials?

Answer 2:

All these parameters are measured, the porosity is related according to the surface density and the thickness. The choice of the material follows from the long term experience of the laboratory and also the medical use.

Changes 2:

NO

Comment 3:

Young-Laplace equation has existed for a long time, and the corresponding calculation of material surface tension also has corresponding instructions. The key is what is the difference between the surfaces of porous materials? What is the novelty of the author's research?

Answer 3:

We are aware that the Young-Laplace eq. is known. We see the novelty of our research is twofold:

- a) Idea of filtration of UHMWPE wear particles that is not described elsewhere.
- b) The wettability of the synovial fluid has not been described in the literature for such specific material as the electrospun membrane is.

Moreover, the known wettability of material in medical use is important not only in the filtration application but also in the drug release research optimization.

Changes 3:

NO.

Comment 4:

As the receiving fabric of electrospinning, what is the basic structure of the fabric, such as specification, thickness, gram weight, plain, twill and non-woven fabric? Fabric structure has a great influence on electrospun membranes.

Answer 4:

Yes, you are right. Nevertheless, the method is described in section Membrane sample preparation. Here is the summary: For the preparation of the electrospun membranes polyamide 6 was used. The solvent system consisted of acetic and formic acids. The polymer solution of PA6 was prepared in a concentration of 12 wt%. The solvent system consisted of the above mentioned acids in a ratio of 2:1. The solution was mixed in the magnetic stirrer under the room temperature for 24 hours until the polymer was fully dissolved. The prepared solution was electrospun by use of Nanospider™ NS 1WS500U with a wire electrode. Different towing velocities (5-20 mm/min) were used to investigate the influence on the contact angle. The electrospinning was done on a polypropylene base Spunbond fabric.

Changes 4:

NO

Comment 5:

In the electrospinning process, the fiber diameter affects the porosity. Does the corresponding surface tension have an effect? Should the method be used to prove?

Answer 5:

Yes, you are right that the porosity is affected by the fibre diameter. The fibre diameter was measured from SEM by Image J software. The corresponding porosity was calculated by eq. 10. The effect can be seen on figure 7.

Changes 5:

NO

Comment 6:

Why only choose the advancing speed of 15 mm/min for spinning? The relationship between fiber diameter and surface tension should be increased. At the same time, the surface morphology after hot pressing and the hot pressing of advancing speed should be improved.

In addition, whether the corresponding pressing temperature is too low, because the melting point of PA6 is obviously greater than 200 °C, how can there be a molten layer without softening, or just hot pressing the non-woven layer, what does it have to do with the electrospinning layer?

Answer 6:

We are sorry, but we have mismatched the images in the original text. The towing velocity of 20 mm/min has resulted in membranes with the lowest FSE values, these membranes were taken for the next lamination process. We have considered this towing value as the worst case scenario and the authors wanted to investigate the effect of lamination on the surface structure and also on the wettability.

If we go over the melting temperature, the membrane would completely close the pores and no filtration process can be performed. That is why we have selected the temperatures and pressure. It comes from our experience with filtration materials.

Changes 6:

The corresponding text and images were changed (from 15 mm/min to 20 mm/min).

Comment 7:

Page 8, Line 12. Is the conclusion wrong? Is there no direct relationship between surface tension and porosity??

Answer 7:

On page 8, line 12 was mentioned: “The difference between the highest and lowest FSE is 6 mN/m, one can notice that with the decreasing porosity the FSE also decreases”. So, there is a relationship between surface tension and porosity. We believe the conclusion is in line with this finding.

Changes 7:

NO

Comment 8:

Why is the electrospun membranes hydrophilic? What is the calculation method of porosity?

Answer 8:

The calculated method for the porosity estimation was based on the surface density, membrane thickness and density of the fibre. The membranes are hydrophilic as they show the contact angle smaller than 90 degree.

Changes 8:

NO

Comment 9:

It is suggested that the author further refine the research work, especially the corresponding characterization data, to increase the persuasion of the paper. It can be considered to focus on the relationship between the structure and properties of the obtained electrospun membranes.

Answer 9:

Thank you for this comment. In our research, we focus mainly on the application of electrospun membranes. Thereby, our main focus is not on the structure improvement but the application of this type of membranes in case of total hip arthroplasty.

Changes 9:

NO